# Screening for Perinatal Anxiety Using the Childbirth Fear Questionnaire: A New Measure of Fear of Childbirth

**DOI:** 10.3390/ijerph19042223

**Published:** 2022-02-16

**Authors:** Nichole Fairbrother, Fanie Collardeau, Arianne Albert, Kathrin Stoll

**Affiliations:** 1Department of Family Practice, Faculty of Medicine, University of British Columbia, Vancouver, BC V6T 1Z4, Canada; kathrin.stoll@ubc.ca; 2Department of Psychology, Faculty of Social Sciences, University of Victoria, Victoria, BC V8P 5C2, Canada; faniecol@uvic.ca; 3Women’s Health Research Institute, Vancouver, BC V6H 2N9, Canada; arianne.albert@cw.bc.ca

**Keywords:** childbirth, fear, assessment, birth, questionnaire development, caesarean, vaginal

## Abstract

Fear of childbirth affects as many as 20% of pregnant people, and has been associated with pregnancy termination, prolonged labour, increased risk of emergency and elective caesarean delivery, poor maternal mental health, and poor maternal-infant bonding. Currently available measures of fear of childbirth fail to fully capture pregnant people’s childbirth-related fears. The purpose of this research was to develop a new measure of fear of childbirth (the Childbirth Fear Questionnaire; CFQ) that would address the limitations of existing measures. The CFQ’s psychometric properties were evaluated through two studies. Participants for Study 1 were 643 pregnant people residing in Canada, the United States, and the United Kingdom, with a mean age of 29.0 (SD = 5.1) years, and 881 pregnant people residing in Canada, with a mean age of 32.9 (SD = 4.3) years for Study 2. In both studies, participants completed a set of questionnaires, including the CFQ, via an online survey. Exploratory factor analysis in Study 1 resulted in a 40-item, 9-factor scale, which was well supported in Study 2. Both studies provided evidence of high internal consistency and convergent and discriminant validity. Study 1 also provided evidence that the CFQ detects group differences between pregnant people across mode of delivery preference and parity. Study 2 added to findings from Study 1 by providing evidence for the dimensional structure of the construct of fear of childbirth, and measurement invariance across parity groups (i.e., the measurement model of the CFQ was generalizable across parity groups). Estimates of the psychometric properties of the CFQ across the two studies provided evidence that the CFQ is psychometrically sound, and currently the most comprehensive measure of fear of childbirth available. The CFQ covers a broad range of domains of fear of childbirth and can serve to identify specific fear domains to be targeted in treatment.

## 1. Introduction

Worldwide, approximately 137 million births occur each year, with over 300,000 babies born in Canada alone [1]. Maternal mortality is lowest in high income countries [2]. Specifically, in 2017 the maternal mortality ratio (MMR: the number of maternal deaths per 100,000 live births, and a measure of the overall quality of maternal health and reproductive care) was 11 per 100,000 live births in high income countries [2]. The MMR is estimated at 10 for Canada and 19 for the United States [3]. In low-income countries, maternal mortality is higher, with 462 death per 100,000 live births [2]. Globally, from 2000 to 2017, the MMR dropped by 38% [2]. However, despite the relative safety of childbirth in developed nations, many pregnant people nevertheless experience high levels of fear of childbirth.

Pregnancy and childbirth are significant, emotionally powerful life events. For many childbearing people, pregnancy follows a complex emotional trajectory characterized by both positive and negative feelings in anticipation of their due date [4,5]. Mental health difficulties are common among perinatal people, with pre- and postnatal depression, and postpartum psychosis the most studied [6].

Until recently, perinatal anxiety had received limited attention [7]. We now know that the anxiety and their related disorder are the most common mental health conditions to affect pregnant and postpartum people [8]. Specifically, one in five pregnant and postpartum people suffer from one or more anxiety or anxiety-related disorders during the perinatal period [8]. This is much greater than the prevalence of depression affecting approximately six to twelve percent of perinatal people [9,10]. The anxiety disorders include all of the core anxiety conditions (generalized anxiety disorder, panic disorder, agoraphobia, social anxiety disorder and specific phobias) as well as obsessive-compulsive disorder and posttraumatic stress disorder. These latter conditions were, until recently also considered anxiety disorders, and many investigators continue to include them among the anxiety and anxiety-related disorders [11]. Among perinatal people, the content of one’s anxiety often orients towards the health and wellbeing of the pregnancy and childbirth (for both the mother and the unborn child), and the health and wellbeing of one’s new-born. For example, worries in generalized anxiety disorder may often involve these areas of concern, and perinatal obsessive-compulsive disorder is often characterized by unwanted, intrusive thoughts of infant-related harm [12,13,14,15,16,17]. A key domain of anxious concern among both nulliparous and multiparous pregnant people is childbirth. Childbirth related fears (e.g., fear of pain, medical interventions, potential harm to one’s infant) are common and can be intense [18].

While positive feelings usually outweigh negative feelings, including worries about childbirth, for some, negative emotions, including fear related to giving birth, predominate (6). Despite the relative safety of childbirth in high income settings, pregnant people may experience fear about being unable to prepare for the unpredictable, the amount of pain they will experience during labour and birth, the possible medical procedures that may be required (e.g., caesarean), as well as concerns for the health and wellbeing of themselves and their new-born (4,6). In a recent, large-scale systematic review and meta-analyses the global pooled prevalence of FoB in pregnant women was estimated at 14% (95% CI 0.12–0.16) [19]. Twenty-nine primary studies conducted in middle- and high-income countries were included in this analysis. Significant between-study heterogeneity was reported, with prevalence estimates ranging from 3.7 to 43%, likely due to variability in methodological quality, measurement tools and cut-scores [19]. All but one [20] of the included studies employed self-report questionnaires as a measure of FoB. When measured using diagnostic interviews, clinically significant levels of fear of childbirth have been found to be much lower. To our knowledge, only three studies have taken this approach, reporting prevalence estimates of 2.4 [21], 4.5 [20], and 8.5% [22] for FoB. This is not surprising, as prevalence estimates are often higher when mental health difficulties are measured using self-report questionnaires, compared to when formal diagnostic criteria are employed [23,24,25]. Of note, studies employing diagnostic criteria to investigate FoB have only been carried out in high-income countries. Therefore, our knowledge of clinically significant levels of FoB in middle- or low-income countries remains limited.

With one exception [20], levels of fear of childbirth have typically been found to be higher among nulliparous people [19,21,26] compared with multiparous people. Fear of childbirth has also been associated with a range of negative outcomes including: avoidance of pregnancy, termination of pregnancy, higher levels of perceived pain during childbirth, increased length of labour, increased likelihood of emergency and elective caesarean birth, postnatal depression and posttraumatic stress disorder, increased parenting stress, and poor maternal-infant bonding [26,27,28,29,30]. There are, however, opportunities to mitigate these negative effects. Previous research has shown that psychotherapy and educational interventions, such as counselling delivered by maternity care providers or education on childbirth at the hospital, can reduce pregnant people’s fears of childbirth [31]. Additionally, although medical indications for caesarean birth have been well-established, to our knowledge, there are few established psychosocial indications. Persistent, untreated fear of childbirth, which is clinically distressing or impairing, may justifiably be one such indication. 

### 1.1. Measurements of Fear of Childbirth

Although numerous measures of childbirth fear have been reported in the literature ranging in length from a single item to 53 items, each of them either: (a) assesses only a subset of the content domains relevant to fear of childbirth [4,32,33,34,35,36,37,38,39,40,41,42]; (b) includes only one or two items for some of the domains assessed [4,43]; (c) includes non-fear-related items [4,32,33,34,35,41,44]; (d) are specific to a particular subpopulation (e.g., adolescents, those who have already given birth) [32,41]; or (e) are single or double item measures only [40,45]. See Table 1 for details.

In our opinion, one or two item measures are insufficient to produce a stable estimate of childbirth fear, nor can they encompass the possible range of concerns experienced by people who are pregnant, or may become pregnant (e.g., fear of harm, medical interventions, or pain). Further, evidence suggests that fear of childbirth is multidimensional [4,44,47]. Among the longer scales developed [4,33,36] each, either fails to assess key domains of childbirth fears (e.g., pain, harm to self or infant), includes non-fear relevant items, or under-samples content domains (i.e., have only one or two items for a particular content domain, whereas a minimum of three is needed to produce a stable measure). 

By far the most commonly used measure of fear of childbirth is the Wijma Delivery Expectancy/Experience Questionnaire Version A (W-DEQ-A) [33]. The W-DEQ-A has been used in several countries [28,48,49,50,51,52,53,54,55,56]. The W-DEQ-A has been found to possess good psychometric properties [28,33,57]. Although psychometrically sound [28,33], the W-DEQ-A is not limited to an assessment of fear, but rather assesses a wide range of perceptions of labour and delivery (e.g., during labour and delivery, do you think you will feel: lonely; strong; confident; afraid; deserted; weak; safe; independent; desolate; tense; happy, etc.). In factor analytic studies of the W-DEQ-A, fear has been found to emerge as one of four, or one of six factors, strongly suggesting that the W-DEQ-A is not only a measure of fear [28,33]. Further, many aspects of fear of childbirth are not addressed in this measure (e.g., pain; perceptions of social embarrassment; pressure to receive/avoid pain medication; mother’s safety; changes to the body and sexual function; fear of medical interventions). The W-DEQ-A may limit researchers’ ability to assess whether specific aspects of fear of childbirth are predicated by different life experiences and/or result in differing outcomes. At the same time, the broad nature of W-DEQ-A items may still capture participants’ experience of fear of childbirth albeit in a general way (i.e., still report fear, even if the exact content of their fear of childbirth would not be known). 

Accurate measurement of fear of childbirth is important in correctly identifying those experiencing high levels of fear of childbirth, as well as identifying targets for treatment. At present, currently available measures of fear of childbirth do not fully meet this standard. 

### 1.2. The Present Studies

We sought to develop a self-report measure of fear of childbirth that recognizes the complexity of childbirth fear and assesses fear of birth regardless of the planned or preferred mode of delivery. The purpose of this research was to evaluate the psychometric properties of a newly developed measure of FoB: Childbirth Fear Questionnaire (CFQ). 

In Study 1, we sought to establish the factor structure of the CFQ (i.e., ascertain the appropriate number of factors and items per factor, and remove items that fail to load sufficiently on any factor). We also sought to conduct a preliminary evaluation of the resulting measure’s reliability and construct validity. In view of these objectives, we conducted an exploratory factor analysis of the initial item pool and identified the items and subscales for Study 2. We also assessed the reliability, and convergent and discriminant validity of the measure. Specifically, we predicted that the CFQ would correlate more strongly with another measure of FoB (W-DEQ-A full scale and the W-DEQ-A fear scale) than with measures of blood and injury fears (the MQ) or depressed mood (the EPDS) [58]. To further assess the construct validity of the CFQ we also compared participants who reported a preference for a vaginal birth to those who reported a preference for a caesarean birth. We predicted that participants who reported a preference for a caesarean birth would also report higher levels of fear of pain from a vaginal birth, fear of harm to baby, fear of mum or baby dying, fear of insufficient pain medication, and fear of damage to one’s body from a vaginal birth, but lower levels of fear of caesarean delivery and fear of medical interventions, compared with those who reported a preference for a vaginal birth.

In Study 2, we evaluated the replicability and generalizability of the factor structure of the CFQ and conducted further reliability, and convergent and discriminant validity evaluations. We tested the convergent/discriminant validity of the CFQ by comparing the relationship between the CFQ and the W-DEQ-A with the relationships between the CFQ and measures of depressed mood (the EPDS) and symptoms of posttraumatic stress disorder (the PDS-5). We predicted that the CFQ would correlate more strongly with both the W-DEQ-A full and fear scale [58] than with either the EPDS or the PDS-5.

## 2. Study 1

### 2.1. Materials and Methods

#### 2.1.1. Participants 

We recruited a convenience sample of English speaking, pregnant people who were over the age of 18 (mean = 29.0 years, SD = 5.1) and who were residing in Canada, the United Kingdom, or the United States to participate, via online forums frequented by pregnant people (e.g., pregnancy-related web sites and blogs). We planned for a sample size of approximately 500 individuals, following the recommendations of MacCallum et al. [59] for the sample size needs of an exploratory factor analysis (EFA). Our final sample consisted of 643 pregnant people.

#### 2.1.2. Procedures 

In order to complete the survey, participants were required to acknowledge that they had read the study cover sheet/consent form and agreed to participate. Consenting participants completed the online survey between 3 and 42 weeks’ gestation. For each survey completed, $0.50 was donated to the Children’s Health Foundation of Vancouver Island, British Columbia. The study was approved by the Behavioural Research Ethics Board of the University of British Columbia.

#### 2.1.3. Measures

Background Questions. Participants completed a set of demographic questions (e.g., age, marital status, education, income, country of residence, and race and ethnicity), questions about the current pregnancy (e.g., method of conception, and number of foetuses), and previous pregnancies (if applicable) (e.g., the number of prior pregnancies, births, miscarriages, and vaginal and caesarean deliveries). 

Birth Preferences. Using a 7-point Likert-type scale (ranging from a very strong preference for a vaginal birth to a very strong preference for a caesarean birth), participants were asked about mode of delivery preference (i.e., vaginal versus caesarean). 

Childbirth Fear Questionnaire (CFQ)—Initial Item Pool. The initial pool of CFQ items and item domains were developed by a team of perinatal researchers from the fields of psychology, midwifery and nursing, and based on earlier work in this area, including our own [60]. Our group of investigators collaborated in reviewing the extant FoB measurement literature, generating items for inclusion in the CFQ, reviewing item wording, and ensuring that all domains of fear deemed relevant to childbirth had been included in the initial pool of items. In recognition of the likelihood that the CFQ would include both a total scale score and subscale scores, we developed multiple items for each fear domain (a minimum of three items per subscale are needed to ensure a reasonable degree of internal consistency reliability). To be able to reduce the overall number of items, and develop subscales with high internal consistency and reliability, each content domain initially included five or more items. 

This process resulted in an initial pool of 49 items covering the following domains of childbirth-related fears: social embarrassment (e.g., fear of losing control), pain (i.e., fear of pain), pain medication (e.g., fear of not receiving the pain medication one is hoping for), mode of delivery (e.g., fear of a caesarean delivery), baby’s and mother’s physical safety (e.g., fear that one’s infant may be harmed or die during labour/delivery), changes to one’s body (e.g., scarring), sexual functioning (e.g., enjoying sexual activity less following delivery), and medical interventions (e.g., fear of having an episiotomy). The CFQ items are scored on a 0 (not at all) to 4 (extremely) point, Likert-type scale.

These initial fear domains represent content areas of fear and concerns commonly reported by pregnant people, such as fear of pain and fear that harm might come to the baby [47,61]. In earlier work [60], maternal complications, feelings of embarrassment, fear of medical interventions/surgery, scarring, sexual functioning, and body damage, were also identified as areas of childbirth related fear and concern, and were considered when developing the initial pool of items for the CFQ. 

Wijma Delivery Expectancy Questionnaire (W-DEQ-A). The W-DEQ-A is a 33-item questionnaire, with items scored on a 0–5 Likert type scale, and a range of possible scores from 0 to 165. The psychometric properties of this assessment tool have been well established [28,33]. In the current sample, the internal consistency reliability for the W-DEQ-A was 0.92. In addition to the W-DEQ-A total score, there are also data to support the use of a 6-item fear scale [58].

In this study, in error, we administered the W-DEQ-A using a 0–4 Likert type scale (rather than the usual 0–5 scale). We then prorated the W-DEQ-A scores to the more standard 0–5 point scale as follows: original W-DEQ-A score was divided by four, then multiplied by 5. Our rescaled mean W-DEQ-A (M = 55.9) score was consistent with those found in the literature, which range from 52.9 to 68.3 [62]. Our mean scores for both nulliparous (60.7) and multiparous people (50.2) were consistent with those reported in the literature (i.e., 54.1 to 68.51 and 50.3 to 60.7, respectively), as was the percentage of participants scoring above 85 (i.e., 7.5% to 15.6% in the literature, and 9.8% in the current study) [62]. We are confident that our prorated W-DEQ-A scores are a valid estimate of correctly scaled W-DEQ-A items and were valid to use as the main measure of convergent validity.

Edinburgh Postnatal Depression Scale (EPDS). The EPDS is a 10-item self-report screening tool for pre and postnatal depression. The sensitivity and specificity of the EPDS are in acceptable ranges (65–100%, and 49–100%, respectively) [63]. The EPDS is the most widely used screening tool for perinatal depression [64]. It was included in this study as a measure of discriminant validity for the CFQ; childbirth fear should be no more than moderately correlated with depression. In the current sample, the internal consistency reliability of the EPDS was 0.88.

Mutilation Questionnaire (MQ). The MQ is a 30-item measure of blood and injury fears. Internal consistency for the MQ ranges from 0.75 to 0.86 [65]. In the current sample, the internal consistency reliability was 0.87. High MQ scores are associated with fainting at the sight of blood and injury [65,66]. The MQ was included as a second measure of discriminant validity for the CFQ; blood-injury fears should be no more than moderately correlated with fear of childbirth. 

#### 2.1.4. Data Analysis Strategy 

Factor analyses were performed in R (v. 3.3.2) [67] using the psych() package (v. 1.6.9) [68] for fitting exploratory factor analysis models. Accompanying visualizations were created using the ggplot2() package (v. 2.2.1) [69]. Differences between correlations were tested using a test provided by Lee and Preacher [70], and standardized mean-difference effect sizes for *t*-tests (ds) were estimated using the calculator provided by Lakens [71]. All other analyses were carried out using IBM SPSS Statistics (v23) (IBM, Chicago, IL, USA). 

Exploratory Factor Analysis. We followed the recommendations of Sakaluk and Short [72] and others (e.g., [73]) for conducting exploratory factor analysis on all CFQ items. Specifically, all solutions extracted common factors via maximum likelihood estimation, to facilitate the calculation of model fit indices. We determined the number of factors to retain through a combination of criteria, including: (1) parallel analysis [74]; (2) the minimum average partial (MAP) criterion [75]; (3) interpretations of absolute and relative indexes of model fit (the root mean square error of approximation [RMSEA], and Tucker-Lewis Index [TLI], respectively) [76]; (4) interpretations of the Bayesian Information Criterion (BIC); (5) nested model comparisons using likelihood-ratio tests between competing models; and (6) factor solution interpretability. All solutions were rotated to achieve simple structure and estimate factor correlations, using the oblique Oblimin method to allow for factor correlation. 

We considered items that loaded onto a factor at ≥0.35 to be substantive indicators of the underlying latent construct. Items that did not load onto any factor beyond this threshold were determined to be poor indicators and were removed from the final version of the CFQ. 

Convergent/Discriminant Validity. The convergent/discriminant validity of the CFQ was assessed via correlation analyses. We compared the correlations between the CFQ and the W-DEQ-A with the correlations between the CFQ and measures of depressed mood (the EPDS) and blood and injury fears (the MQ). We further conducted *t*-tests to compare mean subscale scores between participants with a strong desire for a vaginal birth to participants with a strong desire for a caesarean birth. 

Reliability and Validity Analyses. The remaining analyses involved descriptive data (means, standard deviations, and percentages), Cronbach alpha reliability coefficients, correlations, and independent-samples *t*-tests. Differences between correlations were tested using a test of the difference between two dependent correlations with one variable in common [70]. 

Exploratory Analyses. We also conducted *t*-test analyses to compared CFQ subscale scores across parity and country (Canada and the United States) groups.

### 2.2. Results

#### 2.2.1. Demographics

Participant demographic and reproductive information is presented in Table 2. Note that complications in the current pregnancy were reported by participants (22.9%), and ranged broadly in severity (e.g., early spotting, anaemia, pre-eclampsia). In this sample, 21.3% of participants scored a 12 or greater (common cut-score for depression) on the EPDS.

#### 2.2.2. Exploratory Factor Analysis of the CFQ

Results of our parallel analysis (see Figure 1) suggested a maximum of 13 factors ought to be retained, whereas the MAP test suggested 9 factors was sufficient. We then proceeded to evaluate indexes of model fit, information criteria, and nested model comparisons for 1–13 factor solutions (see Table 3).

Each additionally extracted factor significantly improved the fit of our model. Adequate model fit based on the RMSEA was achieved from a 6-factor model onward, whereas adequate model fit based on the TLI was achieved near the 9- to 10-factor models. The BIC indicated that our models became unnecessarily complex after the 11-factor solution, and our 13-factor solution failed to converge. We therefore examined the pattern matrixes of loadings for the 9-, 10-, and 11-factor solutions. 

The 9-factor solution was supported by the MAP test and had acceptable fit according to the RMSEA, and near-acceptable fit according to the TLI1; it was also the most conceptually interpretable of the three solutions we investigated in detail. The 10-factor solution, though acceptably fitting according to both the RMSEA and TLI, yielded a tenth factor that was not conceptually coherent. The 11-factor solution, finally, was also acceptably fitting, but the extracted eleventh factor had no substantially loading items. We therefore selected the 9-factor solution as the best fitting and conceptually interpretable model of the CFQ items. 

Factor loadings for the final nine factors are presented in Table 4, and represent: (1) Fear of loss of sexual pleasure/attractiveness (SEX), (2) Fear of pain from a vaginal birth (PAIN), (3) Fear of medical interventions (INT), (4) Fear of embarrassment (SHY), (5) Fear of harm to baby (HARM), (6) Fear of caesarean birth (CS), (7) Fear of mum or baby dying (DEATH), (8) Fear of insufficient pain medication (MEDS), and (9) Fear of body damage from a vaginal birth (DAMAGE). Correlations between the nine CFQ factors ranged from weak (r = −0.01) to strong (r = 0.84), with approximately one third equal to or greater than 0.50 (see Table 5 for details). Correlations at or above 0.50 were for: Fear of loss of sexual pleasure/attractiveness with Fear of pain from a vaginal birth, Fear of embarrassment, Fear of insufficient pain medication and Fear of body damage from a vaginal birth; Fear of pain from a vaginal birth with Fear of insufficient pain medication and Fear of body damage from a vaginal birth; Fear of medical interventions with Fear of caesarean birth; Fear of embarrassment with Fear of body damage from a vaginal birth, and Fear of harm to baby with Fear of mom or baby dying and Fear of body damage from a vaginal birth. 

#### 2.2.3. Descriptive, Reliability, and Validity Analyses

Descriptive Analyses. Means and standard deviations for the 9 subscales, and the CFQ Total scale scores are presented in Table 6. 

Reliability Analyses. The Cronbach alpha for the overall 40-item scale was 0.94. Cronbach alphas for the individual subscales ranged from 0.76 to 0.94. Specifically, Cronbach’s alphas were 0.93 for Fear of loss of sexual pleasure/attractiveness, 0.94 for Fear or pain from a vaginal birth, 0.82 for Fear of medical interventions, 0.84 for Fear of embarrassment, 0.93 for Fear of harm to baby, 0.85 for CS, 0.86 for Fear of mum of baby dying, 0.76 for Fear of insufficient pain medication, and 0.85 for Fear of body damage from a vaginal birth. 

Convergent/Discriminant Validity. The correlations between the CFQ and the W-DEQ-A (full and fear scales) were 0.41 (*p* < 0.001) and 0.57 (*p* < 0.001) respectively. The correlation between the CFQ and the EPDS was 0.35 (*p* < 0.001), and the correlation between the CFQ and the MQ was 0.28 (*p* < 0.001). The CFQ-W-DEQ-A (full scale) correlation was significantly greater than the CFQ-MQ correlation, z = 2.73, *p* = 0.006, but not the CFQ-EPDS correlation, z = 1.60, *p* = 0.109. The CFQ-W-DEQ-A (fear scale) correlation was significantly greater than both the CFQ-MQ correlation, z = 7.17, *p* < 0.001, and the CFQ-EPDS correlation, z = 6.61, *p* < 0.001. 

Birth Preferences. Most people in our sample indicated a strong or a very strong preference for a vaginal childbirth (83.8%, *n* = 539), whereas only a small proportion indicated a strong or a very strong preference for a caesarean delivery (5.1%, *n* = 33). Consistent with the above hypotheses, compared with those who strongly preferred a vaginal birth, people who strongly preferred a caesarean delivery reported higher scores on the Fear of pain from a vaginal birth, *t* (34.08) = −2.83, *p* = 0.008, ds = 0.68, Fear of harm to baby, *t* (570) = −2.84, *p* = 0.005, ds = 0.51, Fear of mum or baby dying, *t* (570) = −2.81, *p* = 0.005, ds = 0.50, and Fear of insufficient pain medication, *t* (33.50) = −5.54, *p* < 0.001, ds = 1.53, subscales of the CFQ, but lower scores on the Fear of caesarean birth, *t* (37.22) = 6.64, *p* < 0.001, ds = −1.07, and the Fear of medical interventions, *t* (570) = 2.15, *p* = 0.032, ds = −0.39, subscales of the CFQ. However, our prediction that those who strongly preferred a caesarean birth would report higher scores on the Fear of damage to one’s body from a vaginal birth was not supported, ds = 0.04. The means and standard deviations by mode of delivery preference are presented in Table 7. 

Parity. Nulliparous and multiparous participants differed significantly on six of the nine CFQ subscales, and the CFQ Total scales. In each case, nulliparous participants scored higher than multiparous participants. Specifically, nulliparous participants scored higher than multiparous participants on the following CFQ factors: Fear of loss of sexual pleasure/attractiveness, *t* (639.43) = 6.34, *p* < 0.001, ds = 0.50, Fear of pain from a vaginal birth, *t* (641) = 8.70, *p* < 0.001, ds = 0.69, Fear of embarrassment, *t* (640.80) = 6.29, *p* < 0.001, ds = 0.50, Fear of harm to baby *t* (641) = 2.88, *p* = 0.004, ds = 0.23, Fear of insufficient pain medication *t* (639.22) = 3.98, *p* < 0.001, ds = 0.31, and Fear of body damage from a vaginal birth *t* (641) = 6.49, *p* < 0.001, ds = 0.51, and CFQ Total scores *t* (641) = 5.83, *p* < 0.001, ds = 0.45. Nulliparous and multiparous participants did not differ significantly on the Fear of medical interventions, ds = 0.01, Fear of caesarean birth, ds = 0.13, or the Fear of mum or baby dying subscales, ds = 0.03. Means and standard deviations by parity, are presented in Table 7. 

Country. Canadian and American participants differed on only two of nine CFQ subscales, Fear of medical interventions, *t* (605) = −2.40, *p* = 0.017, ds = 0.21, and Fear of caesarean birth, *t* (605) = −3.00, *p* = 0.003, ds = 0.26. In both instances, American participants reported higher levels of fear, though the magnitude of these nationality differences were generally smaller than those between birth preference and parity groups. Means and standard deviations by country for Canada and the US are presented in Table 7.

#### 2.2.4. Summary

In our initial psychometric evaluation and development study of the initial 49 CFQ items, involving 643 pregnant people, exploratory factor analysis resulted in a 9-factor scale, supported by MAP test with acceptable fit based on RMSEA. The resulting 9 factors represent: (1) Fear of loss of sexual pleasure/attractiveness (SEX), (2) Fear of pain from a vaginal birth (PAIN), (3) Fear of medical interventions (INT), (4) Fear of embarrassment (SHY), (5) Fear of harm to baby (HARM), (6) Fear of caesarean birth (CS), (7) Fear of mum or baby dying (DEATH), (8) Fear of insufficient pain medication (MEDS), and (9) Fear of body damage from a vaginal birth (DAMAGE). Subscales were weakly to moderately correlated with a few strong correlations (Fear of loss of sexual pleasure/attractiveness with Fear of body damage from a vaginal birth, Fear of harm to baby with Fear of mom or baby dying, and Fear of caesarean birth with Fear of medical interventions). Cronbach alpha coefficients for the total scale and individual subscales were all above 0.76, providing evidence of high internal consistency reliability. Strong evidence of convergent/discriminant validity was found when comparing the 9-factor CFQ with another measure of fear of childbirth and measures of blood, injury injection fears and depressed mood. The CFQ subscale means were also compared across subgroups (e.g., preferred mode of delivery) with hypothesized differences supported by the data. 

## 3. Study 2

### 3.1. Methods

#### 3.1.1. Participants 

We recruited a convenience sample of 881 English-speaking, pregnant people living in Canada, and over the age of 18 (mean = 32.9 yrs, SD = 4.3). Participants were located via Facebook and other online forums frequented by pregnant people (e.g., pregnancy-related web sites and blogs). 

#### 3.1.2. Procedures

Consenting participants completed an online survey between 11 and 46 weeks’ gestation with an average of 35 weeks. Participants were eligible to win one of seven $150 prizes. The research was approved by the Behavioural Research Ethics Board of the University of British Columbia.

#### 3.1.3. Measures 

Participants completed an online survey. Similar to Study 1, the online survey included the same background and demographics questions, the 40 CFQ items retained from Study 1, the W-DEQ-A (without the scoring error described in Study 1), and the EPDS. We also administered a measure of PTSD (see below) [77]. The MQ was not administered to this sample. 

Posttraumatic Diagnostic Scale for DSM-5 (PDS-5). The PDS-5 is a self-report tool used to assess post-traumatic stress disorder (PTSD) based on the DSM-5 diagnostic criteria. The PDS-5, one of the most used self-report measures of PTSD, has been found to show good sensitivity and specificity, internal consistency and test-retest reliability, and convergent and discriminant validity [77,78]. A significantly elevated PDS-5 score (i.e., ≥28) yields a sensitivity of 79% and specificity of 78%, allowing for probable prediction of a PTSD diagnosis [77].

#### 3.1.4. Data Analysis Strategy

Confirmatory Factor Analysis and Invariance Testing. We used confirmatory factor analysis (CFA) to test the replicability of our exploratory measurement model from Study 1; we also specified two additional methods factors that we anticipated shared variance on account of the repeated use of the terms “vaginal” (items 8, 19, 20, 31, 35, and 37) and “caesarean” (items 9, 21, and 34). Given the limited number of response options for the CFQ and that our indicators failed to meet assumptions of multivariate normality (Multivariate Skewness *p* < 0.001, Multivariate Kurtosis *p* < 0.001, univariate nonnormality for all indicators *p* < 0.001) for the default maximum-likelihood estimator, we opted instead to use a robust unweighted least squares estimator (ULSM). The model was identified, and the scale of latent variables was set, using a fixed-factor method, whereby latent variances were fixed to a value of 1 and all loadings were freely estimated. We evaluated models using conventional recommended cut offs for absolute and relative indexes of model fit [76,79], including the RMSEA and standardized root mean square residual (SRMR; both recommended to be <0.08), and the TLI and Comparative Fit Index (CFI; both recommended to be >0.90), being mindful of how model reliability can impact the appropriateness of these cut offs (see [80]). We conducted our CFA using the Lavaan() package (v. 0.5-23.1097) [81] for R [82].

Multi-Group Measurement Invariance Testing. We then tested the generalizability of our CFA model by examining measurement invariance across participants based on their experience of pregnancy as primiparous (*n* = 208) or multiparous (*n* = 683) mothers. Establishing measurement invariance is a necessary precursor to group comparisons of factor correlations or means, in order to rule out the possibility that differences from such comparisons simply reflect divergences in the way groups think about the constructs under consideration [79,83]. Specifically, groups must demonstrate the same number of factors and general pattern of loadings (i.e., configural invariance) and factor loadings of comparable magnitude (i.e., weak invariance) for group comparisons of correlations involving the factors to be valid. Moreover, groups must demonstrate configural and weak invariance, alongside intercepts of comparable magnitude (i.e., strong invariance) for group comparisons of factor means to be valid. 

We began the process of testing measurement invariance by fitting and evaluating a configural invariance model. We then used a combination of nested model comparisons and examining the change in model fit indexes to determine whether the constraints imposed by the subsequent levels of invariance (i.e., weak and strong) were supported, e.g., (66); (67). We fitted and evaluated our invariance models using the semTools() package (v. 0.4-14) [84] for R [82], using the same scale-setting, and identification selections from our CFA analysis. 

However, because of convergence issues with the USLM estimator for our invariance testing, we reverted to using a robust maximum-likelihood estimator for specifying invariance models. As a consequence, our invariance models appeared worse fitting than they would have been under the more appropriate USLM estimator (e.g., there was nearly a 0.10 CFI difference between base models depending on estimator selection). We think this compromise is acceptable, given that with these invariance tests we were primarily concerned with relative changes in model fit as we imposed more stringent invariance constraints. 

Taxometric Analyses. Our final analysis regarding the measurement structure of the CFQ involved examining whether—as our factor analysis models presumed—the CFQ was best understood as reflecting some continuous dimension(s) or rather, some number of discrete categories, using taxometric analyses [85] (for reviews see [86,87]). In essence, taxometric analyses function by calculating indexes that ostensibly evidence continuity vs. categorical-ness for a set of observed indicators (e.g., Mean Above Minus Below A Cut, MAMBAC), and then comparing the values of those indexes against those of the same indexes when coming from simulated populations in which a dimensional or categorical structure is specified. Specifically, a comparison curve fit index (CCFI) is computed as the ratio of the degree of misfit for the observed data to a dimensional population compared to a categorical population, with CCFI values less than 0.45 evidencing support for a dimensional model, values greater than 0.55 evidencing support for a categorical model, and values in between indicating an ambiguous outcome. Further, multiple taxometric indexes can be used to compute CCFIs in this fashion; in fact, it is recommended to do so as a form of consistency testing, in order to ensure interpretations are robust to the idiosyncrasies of each index [86]. We therefore evaluated CCFIs from three standardly reported taxometric indexes: MAMBAC, MAXEIG (maximum eigenvalue), and L-MODE (latent mode). 

In order to conduct a taxometric analysis, we had to determine two additional analytic features: the indicators that we would include in the analysis, and the plausible size of a taxon (i.e., the first extracted category) underlying the CFQ, were a categorical solution to be supported [86]. Unlike other forms of latent variable modelling, taxometrics works best when using an efficient (i.e., limited, non-exhaustive) non-redundant (i.e., spanning the conceptual breadth of the construct) set of indicators from a larger measure; in particular Ruscio et al. [88] recommended somewhere between 3–5 indicators (as cited in [87]). As the CFQ contains many more items, we therefore conducted our taxometric analyses three times, using a different sampling of items across the subscales of the CFQ for each stance (Analysis 1: items 2, 8, 34, 39; Analysis 2: items 13, 21, 24, 26, and 40; and Analysis 3: items 3, 7, 9, and 37), which provided us another opportunity to evaluate the consistency of our analyses across different analytic specifications. 

Next, taxometric model fitting requires the specification of a plausible taxon base rate (see [86]) in order to compute the desired CCFI values. As a recently developed alternative to subjectively determining this base rate (e.g., by consulting previous literature, guestimating, etc.) Ruscio et al. [88] developed a method of creating CCFI profiles, in which taxometric analyses were performed iteratively across a range of specified base-rates. The CCFI profile method, though computationally more intensive, is advantageous in that it provides a reliable means of determining whether the underlying measurement model is dimensional or categorical, and, when in fact categorical, CCFI profiles provide the most accurate estimate of the true underlying base rate. We therefore used the CCFI profile method, evaluating of CCFI values (and their average) across the broadest range of taxon base rates (2.5% to 97.5%). 

We conducted all taxometric analyses using the RTaxometrics package [89] in R [82]. 

Descriptive, Reliability, and Validity Analyses. The remaining analyses involved descriptive data (means, standard deviations, and percentages), Cronbach alpha reliability coefficients, and correlations. Differences between correlations were tested using a test of the difference between two dependent correlations with one variable in common [70].

### 3.2. Results

#### 3.2.1. Demographics

Participant demographic and reproductive information is presented in Table 2.

#### 3.2.2. CFA and Invariance of the CFQ

Our CFA of the exploratory measurement model from Study 1 suggested that our model fit the data extremely well, (676) = 4300.63, *p* < 0.001, CFI = 0.977, TLI = 0.974, RMSEA = 0.064 (90% CI: 0.062, 0.066), SRMR = 0.055. Parameter estimates (see Table 8) suggest that our proposed model fit cut-offs were reasonable for detecting model misspecification, as most standardized factor loadings were near or greater than the population values specified in Hu and Bentler’s simulation study [80].

Our measurement invariance analyses, meanwhile, suggested that our measurement model was generalizable across parity status. Model fit was virtually unchanged moving from the non-grouped CFA model to the multi-group configural invariance model, and changes in model fit indexes between the configural invariance and loading invariant models (ΔFI = 0.000, ΔTLI = 0.004, ΔRMSEA = −0.001, ΔSRMR = −0.002, ΔBIC = −204.51) and loading invariant and intercept invariant models (ΔCFI = −0.002, ΔTLI = 0.001, ΔRMSEA = 0.000, ΔSRMR = 0.001, ΔBIC = −127.65) suggested that the added constraints on measurement parameters were reasonable. 

Parity. Using structural equation modelling and comparing against the intercept-invariance model of the CFQ, we again found differences in CFQ scores based on parity, Δ2 (9) = 60.73, *p* < 0.001. As in Study 1, nulliparous participants scored higher than multiparous participants on seven of the nine CFQ factors, including the Fear of loss of sexual pleasure/attractiveness, z = 4.78, *p* < 0.001, ds = 0.52, Fear of pain from a vaginal birth, z = 2.47, *p* = 0.01, ds = 0.19, Fear of medical intervention, z = 2.68, *p* = 0.007, ds = 0.26, Fear of embarrassment, z = 3.77, *p* < 0.001, ds = 0.41, Fear of harm to baby z = 3.67, ds = 0.32, Fear of mom or baby dying, z = 2.27, *p* = 0.02, ds = 0.20, and Fear of body damage, z = 4.50, *p* < 0.011, ds = 0.37. Nulliparous and multiparous participants did not differ significantly on the Fear of caesarean birth, ds = 0.09, Fear of insufficient pain medication, d = 0.05. 

In sum, our CFA-related analyses allow us to infer that the measurement model of the CFQ is both replicable, and generalizable across parity groups, indicating that it is appropriate for use within (and comparisons between) samples of participants who are expecting with different levels of pregnancy experience. 

#### 3.2.3. Taxometric Structure of the CFQ

Our three selected samplings of CFQ items generally exhibited excellent properties for candidate indicators in taxometric analyses, in terms of distributional characteristics, validity coefficients, and within-taxon and within-compliment correlations, see [86,87]. All three CCFI profiles (see Figure 2) strongly supported a dimensional structure for the CFQ and its subscales, as all individual CCFIs (with the exception of one CCFI from in one CCFI profile, MAXEIG in CCFI Profile 2) and their averages, across all three analyses, unambiguously supported dimensional structure (CCFIs < 0.45). We take the consistency of these effects as compelling evidence that the fear of childbirth is best understood as interrelated factors on which individuals differ in degree, not kind [85].

#### 3.2.4. Descriptive, Reliability, and Validity Analyses

Descriptive Analyses. Means and standard deviations for each of the 9 subscales, and the CFQ Total scale scores are presented in Table 6. 

Reliability Analyses. The Cronbach alpha for the overall 40-item scale was 0.94, and the Cronbach alphas for the individual subscales ranged from 0.71 to 0.94 (i.e., Fear of loss of sexual pleasure/attractiveness = 0.90; Fear of pain from a vaginal birth = 0.94; Fear of medical interventions = 0.78; Fear of embarrassment = 0.79; Fear of harm to baby = 0.84; Fear of caesarean birth = 0.87; Fear of mum or baby dying = 0.88; Fear or insufficient pain medication = 0.71; Fear of body damage from a vaginal birth = 0.85). 

Convergent/Discriminant Validity. The correlations between the CFQ and the W-DEQ-A (full and fear scales) were 0.58 (*p* < 0.001) and 0.62 (*p* < 0.001) respectively. The correlation between the CFQ and the EPDS was 0.34 (*p* < 0.001), and the correlation between the CFQ and the PDS-5 was 0.24 (*p* = 0.001). The CFQ-W-DEQ-A (full scale) correlation was significantly greater than the CFQ-EPDS correlation, z = 7.59, *p* < 0.001, and the CFQ-PDS-5 correlation, z = 6.97, *p* < 0.001. The CFQ-W-DEQ-A (fear scale) correlation was significantly greater than both the CFQ-EPDS (z = 9.00, *p* < 0.001), and the CFQ-PDS-5 (z = 7.97, *p* < 0.001) correlations. See Table 5 for a full list of correlations. 

### 3.3. Summary

Study 2 supported the 9-factor structure of the CFQ, and provided evidence of measurement invariance across parity groups. Specifically, those who had previously given birth understood and responded to CFQ items in the same way as participants who had not previously given birth. Additionally, further tests of the CFQ’s latent structure strongly supported a dimensional structure. Thus, fear of childbirth is a construct on which individuals differ in degree rather than in kind (i.e., higher fear of childbirth is not qualitatively different from a lower fear of childbirth). As in Study 1, based on the results from Study 2, it can be inferred that the CFQ demonstrates high reliability, and convergent and discriminant validity. As was true in Study 1 also, overall, nulliparous participants scored higher on CFQ subscales compared to multiparous participants.

## 4. Discussion

The purpose of this research was to develop a new measure of fear of childbirth in pregnant people, that would encompass the breadth of such fears and overcome some of the limitations of commonly used methods to measure them. Exploratory factor analysis of the CFQ resulted in a 40-item, nine-factor questionnaire. Our nine-factor model was supported by a MAP test, exhibited reasonable model fit and good simple structure, and our factors were readily conceptually interpretable. The 9-factor structure of the CFQ was further supported in Study 2, in a larger sample of pregnant participants. Based on psychometric testing across the two studies, we can infer that the CFQ total scale and the nine subscales demonstrated good internal consistency, and convergent and discriminant validity across both studies. 

The taxometric analyses strongly supported a dimensional structure. Thus, fear of childbirth is a construct on which individuals differ in degree rather than in kind (i.e., higher fear of childbirth is not qualitatively different from a lower fear of childbirth). It also suggests that multiple causal influences with small additive effects may best explain more intense fear of childbirth (i.e., rather than a larger single causal factor). While diagnostic categories are frequently used in psychology and might be helpful for clinicians and health authorities to prioritize individuals’ access to treatment, diagnostic categories hypothesize a categorical latent structure. Given the dimensional latent structure of fear of childbirth, it will be important to bear in mind that any cut-off score will be arbitrary and result in a loss of information. It is thus better for future studies to keep the full continuum of scores and respect the dimensional latent structure of the data [90].

Second, our nine-factor model showed initial evidence of measurement invariance between parity groups. Thus, those who had previously given birth understood and responded to CFQ items in the same way as participants who had not previously given birth. Any variations in responses between those two groups will be due to real world differences, rather than to a misspecification of the measurement model for one or the other group. This is especially important, provided our finding that nulliparous participants scored higher than multiparous participants on several of the nine CFQ factors. 

Data from both studies provide excellent support for the convergent and discriminant validity of the CFQ. As predicted, the CFQ correlated most strongly with another measure of fear of childbirth (W-DEQ-A), and less so with measures of depressed mood (EPDS), trauma symptoms (PDS-5) and blood, injury, injection fears (MQ). In addition, correlations with the CFQ were stronger for the W-DEQ-A (fear subscale) compared with the W-DEQ-A (full scale). This was expected and supports our contention that the W-DEQ-A is not strictly a measure of fear (i.e., includes multiple items more relevant to feelings of depressed mood and other positive and negative emotions). Although the W-DEQ-A fear subscale contains six items, only three truly reflect fear (i.e., afraid, tense and panic). The other three (hopelessness, pain and lose control of myself) are not specifically fear items. The weaker correlation between the CFQ and the W-DEQ-A (full scale) provides support for the CFQ as a novel measure of fear of childbirth, with an emphasis on fear, and distinct from the W-DEQ-A. 

Furthermore, the nine factors of the CFQ have the potential to significantly add to our knowledge about fear of childbirth. For example, the nine subscales of the CFQ, identified through factor analysis, make it evident that pregnant people’s concerns about childbirth encompass a broad range of potential fears, and that pregnant people who prefer a caesarean birth have different concerns than those who prefer a vaginal birth. Findings regarding the association of CFQ domains and mode of delivery preferences are consistent with our predictions that those who strongly prefer a caesarean birth are especially fearful of (a) the pain from a vaginal birth, and the possibility that they may not receive sufficient pain medication during labour/delivery, and (b) the possibility that something may go terribly wrong during labour/birth, and they or their infant may be harmed or die. Conversely, pregnant people who strongly prefer a vaginal birth are, as expected, more fearful of caesarean delivery and labour/birth related medical interventions in general. The same pattern of results can also be observed in the intercorrelations among the CFQ subscales. However, in contrast with our predictions, those who strongly prefer a caesarean birth did not report higher levels of fear of damage to one’s body from a vaginal birth was not supported.

In both studies, the lowest CFQ-W-DEQ-A subscale correlation was for the Fear of caesarean birth subscale. The Fear of medical interventions subscale also correlated weakly with the W-DEQ-A in both studies, although less so in study two (i.e., r = 0.33 and 0.38 in study two, and r = 0.10 and 0.03 in study one). This weak relationship between the W-DEQ-A and these two CFQ subscales is likely a function of the fact that the CFQ is unique among measures of fear of childbirth in its assessment of fears related to operational (i.e., caesarean) delivery. 

These finding highlight an interesting phenomenon, not easily assessed by previously available measures of fear of childbirth: that some pregnant people strongly prefer a caesarean birth and are predominantly fearful of the perceived pain and/or danger associated with vaginal delivery, whereas others strongly prefer a vaginal birth and are predominantly fearful of medical interventions in general, and caesarean birth in particular. The link between fear of labour pain and a preference for caesarean delivery is well-documented [60,91]. Very little is known about fears specific to those who prefer vaginal birth. In this regard, the CFQ fills an important gap in our knowledge of the childbirth fears most relevant to pregnant people who prefer a vaginal birth.

The only between-country (Canada and the United States) differences were for the Fear of medical interventions and the Fear of caesarean birth subscales. This is an expected finding as childbirth is a more medicalized experience in the United States in comparison with Canada [92]. Previous research has shown that those who experience pregnancy in a more medicalized birth culture report heightened fear of interventions and other fears that are specific to hospital settings [93]. Consequently, one would expect pregnant people’s fears of medical experiences in childbirth to be heightened. 

Another example, and a novel and important aspect of the CFQ, is the inclusion of subscales measuring (a) pregnant people’s fears about negative, childbirth-related changes to their appearance and sexual functioning, including Fear of loss of sexual pleasure/attractiveness, and (b) Fear of embarrassment because of events occurring during labour/delivery (e.g., fear of urinating in front of others). It is well known that becoming a parent has a significant, and oftentimes negative, impact on one’s romantic relationship, including one’s sexual relationship [94,95]. That this is a concern for pregnant people appears well captured by the CFQ. That fears about a loss of sexual pleasure and attractiveness are associated with a fear of embarrassment during labour/delivery is not surprising in that both involve potential negative judgments by others, and potentially being seen in ways that are perceived as unattractive by typical standards. Our findings demonstrate that fears regarding embarrassment and sexual functioning/appearance are closely related to fears about childbirth pain and bodily damage in the context of a vaginal delivery, as well as harm or death to mum and baby during childbirth. It appears that pregnant people associate pain from a vaginal birth with vaginal damage, and correspondingly with negative changes to their sexual functioning and appearance, and embarrassing aspects of labour/delivery. 

### 4.1. Clinical Implications

Current measures of fear of childbirth fail to assess the full spectrum of perinatal people’s childbirth related fears. Given that fear of childbirth has been associated with several negative medical and social outcomes, an accurate assessment of these fears is important and has implications for pregnant individual’s reproductive and mental health. The development of an effective self-report measure of fear of childbirth will facilitate: (a) the provision of appropriate treatment for those with these fears; (b) assessment of specific aspects of perinatal people’s childbirth related fears; and (c) identification of fear of childbirth as a potential psychosocial indication for a caesarean delivery. The new CFQ will help to identify pregnant people’s specific childbirth concerns, which may be amenable to education or a psychosocial intervention if more extreme.

### 4.2. Limitations

This study is limited by the fact that we collected data from two convenience samples of pregnant people. We did not collect prospective data, nor did we collect data from reproductive-aged people who were not pregnant. A further limitation is the fact that our sample was English-speaking only, highly educated, and predominantly married/common-law, and Caucasian. It is possible that responses to the CFQ may differ by culture, education, and marital status. Until psychometric evaluations of the CFQ have been undertaken in other cultural contexts, generalizability is, limited pregnant people similar to those in the two studies reported here. Finally, online survey administration prevents the calculation of response rates. 

### 4.3. Future Directions

Future research would benefit from an evaluation of the CFQ among reproductive aged people who are not pregnant, but may one day become pregnant or give birth, those who are gender diverse, as well as reproductive-aged people who are biologically male. The attitudes of biologically male people towards birth and fears concerning childbirth have been shown to influence decision-making around mode of delivery [94]. Further, the validity of the measure should be assessed in other cultural contexts beyond predominantly Caucasian, English-speaking countries. 

In our opinion, the most important next steps in the development of the CFQ are to: (a) evaluate the test-retest reliability and sensitivity to change of the CFQ, and (b) assess the CFQ as a screening tool for specific phobia of fear of childbirth (specific phobia is the diagnostic category which has been put forth as the most appropriate classification of fear of childbirth). 

## 5. Conclusions

The Childbirth Fear Questionnaire (CFQ) is a promising new instrument for the multi-factorial assessment of fear of childbirth. Evidence of its reliability and validity has been presented. We hope this new measure proves useful to identify pregnant people with elevated fear of childbirth, and for future research into the fear of childbirth. 

## Figures and Tables

**Figure 1 ijerph-19-02223-f001:**
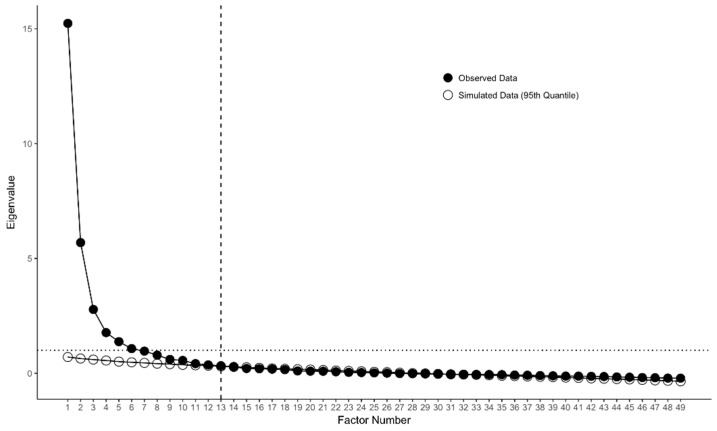
Parallel analysis of 49 CFQ items. Vertical dashed line indicates maximum recommended number of factors (last observed eigenvalue that is larger than 95th quantile of simulated eigenvalues). Horizontal dotted line indicates eigenvalue of 1.

**Figure 2 ijerph-19-02223-f002:**
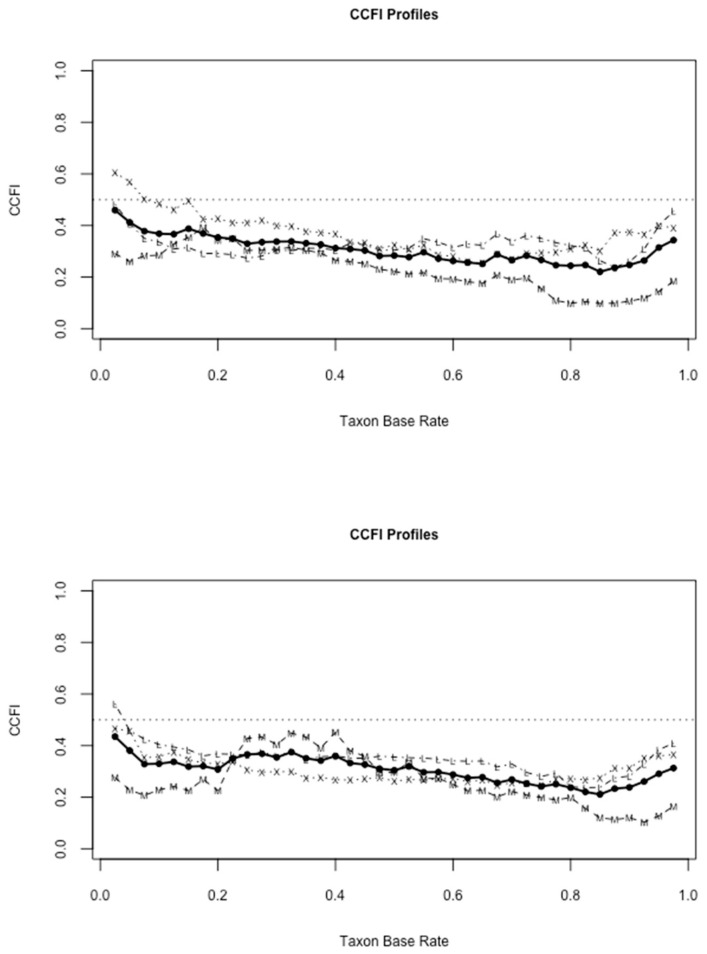
CCFI profiles from taxometric analyses of three unique sets of CFQ indicators. M = MAMBAC, X = MAXEIG, L = L-MODE, and solid dots = the average CCFI.

**Table 1 ijerph-19-02223-t001:** Measurements of fear of childbirth.

Name of Instrument	# of Items	Subscales Include at Least Three Items?	Complete Content Coverage	Excludes Non-Fear Content	For Use with Pregnant People?
Melender (2002)—unnamed [4]	53	NO	MED	NO	YES
Slade-Pais Expectations of Childbirth Scale (SPECS) [44]	50	YES	HIGH	NO	YES
Wijma Delivery Expectancy/Experience Questionnaire—Version A (W-DEQ-A) [33]	33	YES	LOW	NO	YES
Eriksson et al. (2005)—unnamed [34]	29	YES	LOW	NO	NO
Tokophobia Severity Scale (TSS) [46]	13	N/A	MED	NO	YES
Fear of Vaginal Delivery Scale [32]	10	N/A	MED	NO	NO
Birth Experiences Questionnaire (BEQ) [41]	10	N/A	LOW	NO	NO
Oxford Worries about Labour Scale (OWLS) [43]	9	Not all	HIGH	YES	YES
Birth Anticipation Scale (BAS) [36]	6	N/A	LOW	YES	YES
Prelog et al. (2019)—unnamed [42]	6	N/A	-	-	-
Fear of Birth Scale (FOBS) [45]	2	N/A	LOW	YES	YES
Visual Analogue Scale (VAS) [40]	1	N/A	LOW	YES	YES
Hildingsson et al. (2011)—unnamed [38]	1	N/A	LOW	YES	YES
Laursen et al. (2008)—unnamed [39]	1	N/A	LOW	YES	YES

**Table 2 ijerph-19-02223-t002:** Participant demographic and reproductive information. M, mean; SD, standard deviation.

Demographic Variables	Study 1 (*n* = 643)		Study 2 (*n* = 881)	
Age of Participants in Years	18–45 (M = 29.0, SD = 5.1)		19–49 (M = 32.9, SD = 4.3)	
	Percentage	*n*	Percentage	*n*
Married or cohabitating	93.0%	598	93.3%	819
Some postsecondary education	85.9%	520	96.8%	829
Country of residence				
Canada	63.6%	409	100%	881
United States	30.8%	198	N/A	N/A
United Kingdom	2.6%	17	N/A	N/A
European heritage	92.7%	596	72.5%	636
Asian heritage	4.0%	26	10.1%	89
English spoken at home	94.1%	605	94.3%	826
Current pregnancy				
Singleton pregnancy	97.4%	626	97.3%	854
Weeks pregnant: M (SD)	22.2 (10.4)	643	34.9 (2.5)	870
Pregnancy complications	22.9%	147	33.4%	293
Reproductive history				
Prior births	76.3%	*n* = 296	65.9%	*n* = 324
Vaginal	80.7%	239	81.8%	251
Caesarean	25.7%	76	31.0%	85
Prior pregnancy loss < 20 weeks	50.3%	195	39.2%	193
Prior pregnancy loss > 20 weeks	3.9%	15	1.6%	8
Questionnaire data	M	SD	M	SD
W-DEQ-A	55.9	23.5	58.2	23.0
EPDS	7.8	5.1	7.7	5.0
MQ	11.1	6.4	N/A	N/A
PDS-5	N/A	N/A	9.9	11.9

**Table 3 ijerph-19-02223-t003:** Study 1. Model fit indexes, information criteria, and model comparisons for 1–13 factor EFA solutions.

# of Factors	$\chi^{2}$	df	RMSEA	TLI	BIC	$\Delta \chi^{2}$	Δdf
1	14,390.42 ***	1127	0.137	0.405	7103.07	--	--
2	10,789.4 ***	1079	0.116	0.544	3812.43	3601.02 ***	48
3	7965.64 ***	1032	0.104	0.659	1292.58	2823.76 ***	47
4	5796.91 ***	986	0.089	0.752	−578.71	2168.73 ***	46
5	4959.07 ***	941	0.083	0.783	−1125.57	837.84 ***	45
6	4332.69 ***	897	0.079	0.805	−1467.44	626.38 ***	44
7	3712.56 ***	854	0.074	0.830	−1809.53	620.13 ***	43
8	2923.31 ***	812	0.065	0.867	−2327.2	789.25 ***	42
**9**	**2448.91 *****	**771**	**0.060**	**0.889**	**−2536.49**	**474.40 *****	**41**
10	2092.93 ***	731	0.055	0.905	−2633.83	355.98 ***	40
11	1766.84 ***	692	0.051	0.921	−2707.73	326.09 ***	39
12	1549.77 ***	654	0.048	0.930	−2679.09	217.07 ***	38
13 ^a^	1318.40 ***	617	0.044	0.942	−2671.21	231.37 ***	37

^a^ Model did not converge. *** *p* < 0.001.

**Table 4 ijerph-19-02223-t004:** Oblimin-rotated factor loadings from pattern matrix of 9-Factor CFQ solution.

Item Content	F1	F2	F3	F4	F5	F6	F7	F8	F9
Factor 1: Fear of Loss of Sexual Pleasure/Attractiveness (SEX)									
Vaginal stretching from vaginal birth	**0.659**	0.102	−0.020	0.027	0.055	−0.005	0.000	0.032	0.184
Body look less attractive following birth	**0.541**	0.099	−0.042	0.094	−0.004	0.089	−0.058	0.122	0.039
Vagina look less attractive following birth	**0.823**	0.037	−0.018	0.094	−0.048	0.038	−0.002	0.047	−0.003
Enjoying sex less b/c of stretching	**0.894**	−0.026	0.024	−0.032	0.002	−0.068	0.091	−0.055	−0.002
Partner enjoy sex less b/c of stretching	**0.902**	0.020	−0.030	0.065	0.017	0.049	−0.018	−0.009	−0.054
Enjoying sex less b/c of pain	**0.710**	−0.024	0.044	−0.058	0.129	−0.033	0.025	−0.013	0.062
Factor 2: Fear of Pain from a Vaginal Birth (PAIN)									
Experiencing pain during contractions	0.015	**0.957**	0.029	−0.002	0.020	0.010	0.007	0.012	−0.062
Experiencing pain during vaginal birth	0.019	**0.795**	−0.018	0.041	0.016	−0.033	0.020	0.020	0.110
Experiencing pain during labour	0.018	**0.974**	0.032	−0.022	0.021	0.002	−0.005	0.022	−0.030
Experiencing pain pushing baby out	0.002	**0.835**	−0.047	0.046	0.002	−0.003	0.032	0.016	0.085
Having a vaginal birth	−0.063	**0.359**	0.020	0.079	0.017	−0.196	0.041	0.138	0.277
Factor 3: Fear of Medical Interventions (INT)									
Experiencing pain during caesarean birth	−0.003	0.109	**0.580**	−0.049	−0.009	0.148	0.094	0.104	0.013
Harmed because of incompetent care	0.014	−0.116	**0.602**	0.087	0.166	0.048	0.085	−0.022	−0.031
Being left with scars from caesarean birth	0.201	−0.081	**0.451**	−0.105	−0.062	0.214	−0.062	0.163	0.064
Being administered injections	−0.055	0.051	**0.559**	0.146	0.001	0.039	−0.024	−0.169	0.038
Having catheter inserted	0.056	0.050	**0.504**	0.150	0.008	0.069	−0.013	−0.073	0.101
Having general anaesthetic	0.015	−0.004	**0.440**	0.020	0.060	0.187	−0.109	−0.092	0.134
Being administered epidural	−0.014	0.152	**0.446**	0.063	−0.011	0.126	−0.068	−0.319	0.099
Factor 4: Fear of Embarrassment (SHY)									
Being watched by strangers	−0.016	−0.089	0.351	**0.520**	−0.031	−0.015	0.029	0.006	0.034
Losing emotional control	0.167	0.101	0.016	**0.446**	0.071	−0.003	−0.140	0.036	0.066
Others seeing me urinate	0.049	-0.015	0.025	**0.766**	-0.008	-0.008	0.063	0.011	0.052
Others seeing me bowel	0.098	0.071	-0.103	**0.681**	0.066	0.049	0.007	0.085	0.002
Others seeing me naked	0.033	0.067	0.011	**0.755**	0.009	-0.079	0.038	0.045	-0.025
Factor 5: Fear of Harm to Baby (HARM)									
Baby being harmed during labour/birth	0.001	0.058	-0.039	0.013	**0.924**	-0.015	0.044	0.018	-0.001
Baby being damaged during labour/birth	0.058	0.041	−0.062	0.011	**0.886**	−0.002	0.046	0.010	−0.008
Baby being hurt by medical intervention	−0.014	−0.089	0.249	−0.011	**0.654**	0.095	0.108	0.012	0.055
Factor 6: Fear of Caesarean Birth (CS)									
Not being able to have birth I want	0.002	−0.052	−0.025	0.028	0.077	**0.792**	0.015	0.116	0.043
Not being able to have vaginal birth	0.004	−0.008	−0.038	−0.036	0.001	**0.932**	0.014	−0.034	−0.021
Having a caesarean birth	0.021	0.104	0.331	−0.046	−0.084	**0.581**	0.056	−0.034	0.000
Factor 7: Fear of Baby or Mum Dying (DEATH)									
Baby dying during labour/birth	0.000	0.030	0.000	0.006	0.071	0.025	**0.899**	−0.012	−0.013
Baby suffocating during labour/birth	0.013	0.001	−0.037	0.005	−0.001	0.006	**0.955**	0.015	0.041
Dying during labour/birth	0.129	0.007	0.277	0.072	0.089	−0.048	**0.377**	0.146	−0.135
Factor 8: Fear of Insufficient Pain Medication (MEDS)									
Not getting needed pain meds	0.016	0.097	0.044	0.078	0.048	0.033	0.033	**0.794**	0.002
Not having epidural during labour	0.000	0.059	−0.027	0.040	0.026	0.027	0.043	**0.790**	0.050
Not being able to have c-section	0.033	−0.093	0.030	0.083	0.068	−0.149	0.030	**0.297**	0.201
Factor 9: Fear of Body Damage (DAMAGE)									
Vaginal tearing during birth	0.107	0.135	−0.150	0.044	0.053	0.010	0.074	0.028	**0.685**
Rectal tearing during birth	0.061	0.025	0.011	0.140	0.151	0.005	0.112	0.065	**0.553**
Having an episiotomy	0.012	0.083	0.263	−0.002	−0.024	0.075	0.024	−0.012	**0.496**
Requiring vacuum or forceps	−0.008	−0.019	0.220	0.020	0.297	0.155	−0.045	0.035	**0.394**
Needing stitches	0.207	0.122	0.057	0.068	−0.078	0.037	0.130	0.028	**0.483**
Discarded Items									
Not being strong	0.109	0.182	−0.112	0.340	0.129	0.241	−0.044	0.043	0.024
Receiving unwanted pain meds	−0.007	−0.115	0.322	0.164	0.121	0.294	0.024	−0.271	0.119
Feeling pressure to receive pain meds	−0.053	−0.008	0.217	0.159	−0.004	0.332	0.096	−0.314	0.106
Feeling pressure to have natural birth	0.005	0.149	−0.044	0.185	−0.043	0.033	0.121	0.289	−0.036
Baby contract illness during labour/birth	0.104	−0.065	0.088	−0.015	0.195	−0.055	0.338	0.118	0.039
Bleeding too much during labour/birth	0.156	0.066	0.258	0.105	0.155	−0.087	0.177	0.121	−0.047
Being left with scars from vaginal birth	0.323	0.047	0.192	−0.096	0.090	−0.124	−0.056	0.209	0.334
Having scars/wounds not healing	0.218	−0.029	0.238	0.104	0.116	0.060	0.049	0.168	0.153
Vomiting during labour/birth	0.050	0.139	−0.005	0.239	−0.012	−0.028	0.170	−0.022	0.184

Bold: the factor loadings of each item belonging to each subscale.

**Table 5 ijerph-19-02223-t005:** Correlations among the CFQ subscales and total score for Study 1 (S1) and Study 2 (S2).

Fear of…	SEX		PAIN		INT		SHY		HARM		CS		DEATH		MEDS		DAMAGE	
	S1	S2	S1	S2	S1	S2	S1	S2	S1	S2	S1	S2	S1	S2	S1	S2	S1	S2
1. SEX	--																	
2. PAIN	0.53 **	0.48 **	--															
3. INT	0.22 **	0.34 **	0.08 **	0.34 **	--													
4. SHY	0.59 **	0.51 **	0.49 **	0.50 **	0.34 **	0.40 **	--											
5. HARM	0.49 **	0.38 **	0.33 **	0.38 **	0.30 **	0.50 **	0.40 **	0.35 **	--									
6. CS	0.13 *	0.11 **	−0.01	0.11 *	0.60 **	0.59 **	0.16 **	0.18 **	0.27 **	0.31 **	--							
7. DEATH	0.45 **	0.38 **	0.35 **	0.38 **	0.23 **	0.45 **	0.42 **	0.36 **	0.78 **	0.79 **	0.20 **	0.22 **	--					
8. MEDS	0.50 **	0.37 **	0.59 **	0.62 **	−0.03	0.25 **	0.40 **	0.41 **	0.43 **	0.36 **	−0.04	−0.03	0.43 **	0.35 **	--			
9. DAMAGE	0.62 **	0.55 **	0.56 **	0.64 **	0.45 **	0.54 **	0.54 **	0.51 **	0.57 **	0.59 **	0.31 **	0.34 **	0.47 **	0.50 **	0.42 **	0.47 **	--	
Total	0.78 **	0.69 **	0.66 **	0.74 **	0.58 **	0.74 **	0.73 **	0.68 **	0.73 **	0.73 **	0.43 **	0.46 **	0.68 **	0.69 **	0.56 **	0.58 **	0.84 **	0.84 **

Note. ** = *p* < 0.001; * = *p* < 0.01.

**Table 6 ijerph-19-02223-t006:** CFQ total and subscale means (M) and standard deviations (SD).

Subscale	Study 1 (*n* = 643)	Study 2 (*n* = 874)
Fear of…	M (SD)	M (SD)
loss of sexual pleasure/attractiveness (SEX)	1.20 (1.10)	0.82 (0.78)
pain from a vaginal birth (PAIN)	1.65 (1.10)	1.36 (1.02)
medical interventions (INT)	1.72 (.97)	1.05 (0.74)
embarrassment (SHY)	1.15 (0.95)	0.64 (0.66)
harm to baby (HARM)	2.25 (1.32)	1.57 (1.04)
caesarean birth (CS)	2.29 (1.23)	1.69 (1.09)
mum or baby dying (DEATH)	1.68 (1.30)	1.34 (1.14)
insufficient pain medication (MEDS)	0.80 (0.94)	0.60 (0.76)
body damage from a vaginal birth (DAMAGE)	1.89 (1.05)	1.52 (0.89)
CFQ Total Mean Scores	1.59 (0.73)	1.14 (0.61)

Note. All scores are mean item scores with a possible range of 0 (not at all) to 4 (extremely).

**Table 7 ijerph-19-02223-t007:** CFQ subscale means (M) and standard deviations (SD): by delivery preference and country.

Subscale	Birth Preferences		Parity		Nationality	
	Vaginal (*n* = 539)	Caesarean (*n* = 33)	Nulliparous (*n* = 347)	Multiparous (*n* = 296)	Canada (*n* = 409)	USA (*n* = 198)
Fear of…	M (SD)	M (SD)	M (SD)	M (SD)	M (SD)	M (SD)
loss of sexual pleasure/attractiveness (SEX)	1.1 (1.1)	1.4 (1.3)	1.4 (1.10)	0.9 (1.0) ***	1.2 (1.1)	1.2 (1.1)
pain from a vaginal birth (PAIN)	1.5 (1.0)	2.3 (1.4) **	2.0 (1.1)	1.3 (1.0) ***	1.6 (1.1)	1.7 (1.1)
medical interventions (INT)	1.8 (1.0)	1.4 (0.9) *	1.7 (0.9)	1.7 (1.0)	1.7 (1.0)	1.9 (1.0) *
embarrassment (SHY)	1.1 (0.9)	1.2 (1.1)	1.4 (1.0)	0.9 (0.9) ***	1.1 (0.9)	1.2 (.9)
harm to baby (HARM)	2.2 (1.3)	2.9 (1.5) **	2.4 (1.3)	2.1 (1.3) **	2.3 (1.3)	2.1 (1.3)
caesarean birth (CS)	2.5 (1.1)	1.3 (1.0) ***	2.2 (1.2)	2.4 (1.26)	2.2 (1.2)	2.5 (1.2) **
mum or baby dying (DEATH)	1.6 (1.3)	2.3 (1.2) **	1.7 (1.3)	1.7 (1.3)	1.6 (1.3)	1.8 (1.3)
insufficient pain medication (MEDS)	0.7 (0.8)	2.0 (1.4) ***	0.9 (1.0)	0.6 (0.9) ***	0.8 (0.9)	0.8 (1.0)
body damage from a vaginal birth (DAMAGE)	1.9 (1.0)	1.9 (1.5)	2.1 (1.0)	1.6 (1.0) ***	1.9 (1.1)	1.8 (1.0)
CFQ Total Mean Scores	1.6 (0.7)	1.8 (1.0)	1.7 (0.7)	1.4 (0.7) ***	1.6 (0.7)	1.6 (.7)

Note. All scores are mean item scores with a possible range of 0 (not at all) to 4 (extremely). * *p* < 0.05; ** *p* < 0.01; *** *p* < 0.001, based on *t*-tests for independent samples comparing women who: (a) prefer a vaginal birth to those who prefer a caesarean birth, (b) are nulliparous to women who are multiparous, and (c) are resident of Canada to those who are resident of the United States of America.

**Table 8 ijerph-19-02223-t008:** Model fit indexes for CFA model and invariance between nationality (Canada vs. USA) and parity (primiparous vs. multiparous) groups.

Model	$\chi^{2}$	df	CFI	TLI	RMSEA 90% CI	SRMR	$\Delta \chi^{2}\;$	Δdf	ΔCFI
CFA	1981.26 ***	683	0.93	0.92	0.056, 0.062	0.05	--	--	--
Invariance (Nationality)									
Configural	2799.21 ***	1366	0.92	0.91	0.059, 0.065	0.06	--	--	--
Weak	2874.15 ***	1418	0.92	0.91	0.058, 0.065	0.06	76.69 *	52	0.001
Strong	2927.91 ***	1449	0.92	0.91	0.055, 0.061	0.06	52.65 **	31	0.001
Invariance (Parity)									
Configural	2790.74 ***	1366	0.91	0.90	0.059, 0.065	0.06	--	--	--
Weak	2864.79 ***	1418	0.91	0.90	0.058, 0.065	0.06	76.67 *	52	0.001
Strong	2943.60 ***	1449	0.91	0.90	0.058, 0.065	0.06	80.08 ***	31	0.003

Note. $\chi^{2}$ is Yuan-Bentler corrected version, based on robust MLR estimation; $\Delta \chi^{2}$ is therefore computed using scaled Satorra-Bentler (2001) method. * *p* < 0.05; ** *p* < 0.01; *** *p* < 0.001

## Data Availability

The datasets used and/or analysed during the current study are available from the corresponding author on reasonable request.

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
