# Peer review of "Screening for Perinatal Anxiety Using the Childbirth Fear Questionnaire: A New Measure of Fear of Childbirth"

_ijerph, 2022, doi:10.3390/ijerph19042223_

Round 1

Reviewer 1 Report

The present manuscript (ms.) is well written, organized and structured. Statistical analyzes are adequate with respect to the objectives of this study. In any case, this ms. presents some theoretical and methodological-statistical limitations, so this work cannot be published in its current version.

Throughout the manuscript it is suggested that attention be paid to the language used to discuss reliability and validity issues. Specifically, it is suggested that the language be adjusted to reflect current language for validity evidence. For example, tests are not valid and reliable. Instead, evidence is gathered to support the inferences drawn from the scores (e.g., AERA, APA, & NCME, 1999).

American Educational Research Association, American Psychological Association, & National Council on Measurement in Education (1999). Standards for educational and psychological testing. Washington, DC: American Educational Research Association.

Also, authors should write their ms. complying with the rules of the following work:

Bartram, D., & Hambleton, R. K. (2016). The ITC guidelines: International standards and guidelines relating to tests and testing. In F. T. L. Leong, D. Bartram, F. M. Cheung, K. F. Geisinger, & D. Iliescu (Eds.), The ITC international handbook of testing and assessment (pp. 35–46). Oxford University Press. https://doi.org/10.1093/med:psych/9780199356942.003.0004

ABSTRACT

The authors state the following:

Participants were 643 pregnant people residing in English speaking countries for study one, and 881 pregnant people residing in Canada for study two.

This sentence is confusing. Which English speaking countries are the authors referring to? Canada is also an English speaking country (English is an official co-language). So why this distinction between countries and languages? I do not understand.

Authors should indicate the exact type of sampling performed in both studies.

The authors should indicate the age range, mean and SD for the age variable in both studies.

The previously stated concerns should also be described in the PARTICIPANTS section.

Authors should specify the specific type of exploratory factor analysis performed in this study (e.g., principal components or iterated principal axes-PAF).

INTRODUCTION

This section should end by specifying clearly and specifically what the main objective of this ms. is, as well as the specific objectives and their corresponding hypotheses (one for each specific objective). Of course, these hypotheses must be supported bearing in mind the previous evidence, not the results found in this study. Furthermore, these hypotheses must be robustly accepted or rejected in the DISCUSSION section.

PARTICIPANTS

See my comments for the Abstract section.

How were outliers and missing data handled statistically?

MEASURES

CFQ

Why was the initial CFQ item bank composed of 49 items? Provide robust reasons for this concern.

Accurately describe the characteristics of the expert judges (i.e.,% men or women, mean years of experience, qualifications, etc., etc.).

Were the expert judges authors of this research or external personnel? The authors should clarify this concern.

What were the theoretical-statistical criteria followed by the expert judges?

What was the inter-judge agreement rate?

Was the counterbalance technique applied in the administration of this evaluation battery?

STATISTICAL ANALYS

Specify type of factor analysis performed. Why oblimin rotation? This concern must be justified.

RESULTS

The selection of the 9-factor model for the CFQ is not correctly justified (see Table 2) bearing in mind the similarity of the goodness-of-fit indices between the models examined. More robust reasons are clearly necessary.

Table 5. The correlation coefficients presented in this table must be correctly described (Results section) and correctly interpreted (Diction section) according to the classification proposed by Cohen (1988) regarding the effect sizes for Pearson's correlation coefficients.

The rest of the effect sizes presented in the Results section must be correctly interpreted in the Discussion section. Otherwise, the authors may reach uncertain, safe and even pretentious conclusions.

DISCUSSION

This section should be totally rewritten.

Reviewer 2 Report

The authors have addressed a relevant topic that concerns many women during pregnancy and therefore may induce a negative impact on both fetal and pediatric growth and (neuro)development. I agree with the authors that for validity purposes this questionnaire as a potential measuring tool for childbirth fear should be extrapolated to other settings of populations with varying cultural background.

The manuscript is clearly written. Statistical methods used for the analyses are extensive and appropriate.

Background:

  1. “Clinically significant levels of fear of childbirth have been found to affect as many as 4.5% of pregnant people (7,8).” (lines 57-58). Is this a global percentage or for high-income countries only? Please quote literature on fear of childbirth in middle-income or low-income countries, if existing.

Methods:

  1. What was the range of the gestational age of the participants included in study 1?

Results:

  1. The authors do mention this in the limitation section, however it would be helpful to present the ethnic distribution of the participants in the demographic table, since that would indicate the (non)-applicability of the CFQ questionnaire in similar settings.

Discussion:

  1. Could the level of education and the fact that most participants had postsecondary education have influenced their responses on the CFQ questionnaire, and could this be significantly different from those who were single/unmarried and/or had a lower educational level? Please add a comment on this in the discussion

Reviewer 3 Report

This is a well-written and well-conducted scale development study to measure fear of childbirth. This is a good study in general. My evaluation on this paper is very positive. The data analysis was done carefully and appropriately. I only have a few comments to improve the study further:

  1. Table 1 can be revised further to improve its readability. For example, for those scales that have no name, it will be good to include the authors' name so that the readers can differentiate the scales better within the table.
  2. The authors should include the justifications/criteria on how the model fits are interpreted in Study 1
  3. I would appreciate if the material and data is made open access (e.g. on the Open Science Framework) as this will facilitate meta-analysis (which the authors benefit from too)
  4. Although the item contents are described in the Table 4, it will be helpful for the readers if the authors include the full questionnaire (including the Likert scale) in the appendix.

Round 2

Reviewer 1 Report

I have reviewed the authors's responses to my comments and suggestions. The authors have made a great effort to improve the quality and scientific rigor of this ms.

Therefore, the current version of the ms. could be published in IJRECH.

Reviewer 3 Report

The authors have sufficiently addressed my comments.